# Synergetic Effect of Potassium Oxysalts on Combustion and Ignition of Al/CuO Composites

**DOI:** 10.3390/nano11123366

**Published:** 2021-12-12

**Authors:** Xiaohang Ma, Wanjun Zhao, Wei Le, Jianxin Li, Pengwan Chen, Qingjie Jiao

**Affiliations:** State Key Laboratory of Explosion Science and Technology, Beijing Institute of Technology, Beijing 100081, China; mmxhang@163.com (X.M.); 3120190196@bit.edu.cn (W.L.); Ljx19950204@126.com (J.L.); pwchen@bit.edu.cn (P.C.); jqj@bit.edu.cn (Q.J.)

**Keywords:** potassium oxysalts, synergetic effect, combustion performance, copper oxide, reactivity, ignition

## Abstract

In this study, we studied the synergetic effect of potassium oxysalts on combustion and ignition of nano aluminum (Al) and nano copper oxide (CuO) composites. Potassium periodate (KIO_4_) and potassium perchlorate (KClO_4_) are good oxidizers with high oxygen content and strong oxidizability. Different contents of KIO_4_ and KClO_4_ were added to nano Al/CuO and the composites were assembled by sonication. When the peak pressure of nano Al/CuO was increased ~5–13 times, the pressurization rate was improved by ~1–3 orders of magnitude, the ignition delay time was shortened by ~0.08 ms–0.52 ms and the reaction completeness was adjustable when 30–70% KIO_4_ and KClO_4_ were added into the composites. The reaction of Al/KIO_4_ and Al/KClO_4_ at a lower temperature was helpful to ignite the ternary composite. Meanwhile, CuO significantly reduced the peak temperature of oxygen released from the decomposition of KIO_4_ and KClO_4_. The synergetic effect of binary oxidizers made the combustion performance of the ternary composites better than that of the binary composites. The present work indicates that KIO_4_ and KClO_4_ are promising additives for nano Al/CuO to tune and promote the combustion performance. The ternary composites have potential application in energy devices and combustion apparatus.

## 1. Introduction

Metastable intermolecular composites (MIC) are a kind of energetic material composed of nano-sized fuels and oxidizers. With the development and application of nanotechnology, MIC have recently received more and more attention in the field of energetic materials. The ultra-refinement or nanocrystallization of the energetic materials can significantly improve the energy release and increase the reaction rate [1,2,3,4]. Nano aluminum (Al) powder has become the most commonly used metallic fuel in the MIC system due to its high energy release and low cost [5,6]. The performances of binary MIC systems composed of Al and metallic oxides (such as iron oxide (Fe_2_O_3_), copper oxide (CuO), tungsten trioxide (WO_3_), nickel oxide (NiO), molybdenum trioxide (MoO_3_), cobalt oxide (Co_3_O_4_), and bismuth trioxide (Bi_2_O_3_)) have been prepared through different methods and studied by many scholars at home and abroad [7,8,9,10,11,12,13,14,15,16,17].

However, due to the low oxygen content in common metallic oxides, MIC with these metallic oxidizers as the single oxidizer possess less gas production and slower reaction rate. In order to solve this problem, suitable double oxidizers are introduced to replace the single oxidizers in the MIC system to form a ternary system, or ammonium perchlorate (NH_4_ClO_4_, AP), potassium perchlorate (KClO_4_), potassium periodate (KIO_4_), potassium permanganate (KMnO_4_), etc., with high oxygen content and strong oxidizability, are also used to replace traditional metallic oxidizers, both of which can increase the reaction rate of MIC. Huang found that the binary mixture of Bi_2_O_3_ and CuO was more effective than any single oxide in improving the combustion performance of B [18]. The reaction rate of Al/Fe_2_O_3_/WO_3_ ternary MIC system prepared by Zachariah was dramatically higher than that of any single system [19]. Prakash found that the reaction rate of Al/KMnO_4_ was much higher than that of Al/Fe_2_O_3_ due to the high concentration of the free oxygen [20]. The ignition temperature of the Al/KIO_4_ and Al/sodium periodate (NaIO_4_) binary system prepared by Jian was lower than that of Al/CuO, and its peak pressure and pressurization rate were much higher than that of Al/CuO [21]. Song added a small amount of KClO_4_ and found that its activation energy was lower than that of Al/MnO_2_ and its combustion rate increased [22]. Wang found that the Al/CuO/AP composite with 13% AP had the best combustion performance, and its peak pressure and pressurization rate were three times higher than the traditional Al/CuO nano-thermites [23]. Thus, potassium oxysalts and double oxidizers play a significant role in improving the combustion and ignition performance of the thermites.

Herein, the effects of KIO_4_ and KClO_4_ on the combustion behavior of Al/CuO were systematically studied. A scanning electron microscope (SEM) was used to characterize the prepared energetic composite particles. The theoretical adiabatic flame temperature (AFT) of the energetic composite material was calculated by REAL software. The thermal decomposition reaction behavior of CuO/xKIO_4_ and CuO/xKIO_4_ was studied by differential scanning calorimetry (DSC) and thermogravimetric analysis (TG). The peak pressure and pressurization rate of the MIC were characterized by the combustion test. A carbon dioxide (CO_2_) laser igniter and a high-speed camera were used to test and record the ignition delay time of the composite material. The calorimeter was used to measure the heat release of Al/CuO/xKIO_4_ and Al/CuO/xKClO_4_ composites. The results show that KIO_4_ and KClO_4_ play important roles in enhancing the combustion performance of nano Al-based MIC composites.

## 2. Experimental Section

### 2.1. Chemicals and Preparation of Composites

Aluminum nanoparticles (~30–50 nm) and copper oxide nanoparticles (<50 nm) were purchased from Aladdin Industrial Corporation (Shanghai, China), and KIO_4_ and KClO_4_ were purchased from Sinopharm Chemical Reagent Corporation (Shanghai, China), they were ground in the agate mortar before being used in the experiment. The formulations of composites are shown in Table 1, which are based on Equations (1)–(3). The active metals were considered as the fuel with the oxide layer thickness of ~2–5 nm for nAl [24]. Al/CuO/30% KClO_4_ means that the molar percentage of KClO_4_ and CuO in the oxidizers are 30% and 70%, respectively. The content of Al was stoichiometric assuming the fuel reacted with the oxidizer completely. The samples were prepared via sonication method. The weighted Al and oxidizers were put into a vial with ~10 mL hexane, followed by sonication of ~30 min, and dried in a hood for 24 h. Finally, the dried powders could be obtained for further measurements.
2Al + 3CuO → Al_2_O_3_ + 3Cu, ∆H = −4130 J g^−1^
(1)
8Al + 3KIO_4_ → 4Al_2_O_3_ + 3KI, ∆H = −6016 J g^−1^
(2)
8Al + 3KClO_4_ → 4Al_2_O_3_ + 3KCl, ∆H = −10659 J g^−1^
(3)

### 2.2. Characterization

Scanning electron microscope (SEM, Hitachi S-4800, Tokyo, Japan) was used to analyze the morphologies of the raw materials and composites. Thermogravimetric analysis and differential scanning calorimetry (TG-DSC, Netzsch STA 449 F3, Selb, Germany) were used to characterize the thermodynamic behavior of CuO/xKIO_4_ and CuO/xKClO_4_ binary systems and potassium oxysalts. The sample was heated from room temperature to 700 °C in an argon atmosphere (50 mL min^−1^) at a heating rate of 10 °C min^−1^.

### 2.3. Pressure Cell Tests

The peak pressure was evaluated by combustion cell. Around 25 mg-samples were weighted and loaded into a combustion cell with a constant volume of ~20 cm^3^. The samples were ignited via joule heating through a nichrome wire above the samples, which was connected to a voltage supply. When ignited, the changes of pressure in time were recorded electrically. The pressurization rate was obtained by calculating the initial slope of the pressure rise, which has been used to present the reactivity of energetic materials. A detailed description about pressure cell could be found in the reference [25].

### 2.4. Characterization of Ignition and Combustion

Typically, ~15 mg samples were weighted and placed in the center of the specimen stage and the ignition performance of the samples was evaluated by a CO_2_ laser of 60 W in air. Ignition delay was defined as the time length from the start of the laser to the initial visible spark spot captured by a high-speed camera (i-SPEED 726, Rochford, UK) at 50,000 fps.

### 2.5. Heat Release Measurement

The released heat by energetic materials has been evaluated via a calorimeter [26].

The diagrammatic sketch of the calorimeter is shown in Appendix A. Around 200 mg of thermites with a nichrome wire above were placed into the steel crucible inside the calorimeter. The calorimeter was sealed, then thermites were ignited by the nichrome wire via joule heating in air. The heat released by energetic composites dispersed into the water bath within the calorimeter and caused a raise of temperature. Thus, the combustion heat of thermites could be measured. Every measurement was conducted in triplicate, and the average values were shown accompanied with the standard deviation.

## 3. Results and Discussion

### 3.1. REAL Calculation Results

Based on the formulations of composites, assuming the reaction between the fuel and oxidizers occurs in a constant volume, the REAL code was used based on the principle of minimum free energy to calculate the influence of the addition of KIO_4_ and KClO_4_ on the adiabatic flame temperature (AFT) of the Al/CuO system. As demonstrated in Figure 1, the addition of KIO_4_ and KClO_4_ can enhance the AFT of Al/CuO significantly. The calculated AFT increase monotonously with the increasing molar percentages of potassium oxysalts. Figure 1 demonstrates that the addition of 50% KIO_4_ and 50% KClO_4_ can raise the AFT by ~900 °C and ~1100 °C, respectively. Consequently, the addition of potassium oxysalts could improve the AFT of Al/CuO theoretically. In the reaction process, the addition of potassium salts can significantly increase the adiabatic flame temperature of the system, thus promoting the decomposition of the oxidizers, and enabling the Al and oxidizers to react more completely [19].

### 3.2. Morphology Characterization

The morphologies of the raw materials, Al/CuO/xKIO_4_ and Al/CuO/xKClO_4_ composites were characterized via SEM. As shown in Figure 2, the CuO and Al nanoparticles adhere to micron-sized KIO_4_ and KClO_4_ evenly, indicating that the fuels and oxidizers could be mixed up well via sonication.

### 3.3. Reactivity Characterization

As shown in Figure 3, with the incorporation of KIO_4_ and KClO_4_, the combustion performance of Al/CuO was improved significantly. Among the Al/CuO/potassium oxysalts ternary systems, the highest combustion performance, including the highest peak pressure and pressurization rate, was achieved by Al/CuO/30% potassium oxysalts.

As shown in Figure 3a, the peak pressure of all the ternary systems is higher than that of the binary composites. The peak pressure of Al/CuO/30% KClO_4_ is 10 times and 4 times higher than that of Al/CuO and Al/KClO_4_, respectively. With the increase of the content of KClO_4_, the peak pressure reduced slightly compared with Al/CuO/30% KClO_4_. The peak pressures of Al/CuO/50% KClO_4_ and Al/CuO/70% KClO_4_ are still higher than that of Al/CuO and Al/KClO_4_, which are 8 times and 5 times higher than that of Al/CuO, respectively. Similarly, the peak pressure of Al/CuO/30% KIO_4_ is 13 times and 8 times higher than Al/CuO and Al/KIO_4_, respectively. Therefore, the peak pressure of Al/CuO/30% KIO_4_ (49600 KPa/g) is the highest among all the ternary systems.

For the pressurization rate, which refers to the reactivity of energetic materials [5], the trend is the same as that of peak pressure (Figure 3b). The reactivity of the ternary composites of Al/CuO/xKIO_4_ and Al/CuO/xKClO_4_ is higher than that of conventional Al/KIO_4_, Al/KClO_4_, and Al/CuO binary composites. In the meanwhile, the reactivity of Al/CuO/30% potassium oxysalts is the highest. With the addition of 30% KIO_4_, the pressurization rate is tremendously raised, which is three orders of magnitude higher than Al/CuO and Al/KIO_4_, which is the highest pressurization rate in this study. With the increasing content of KIO_4_, the pressurization rate reduced slightly. The pressurization rates of Al/CuO/50% KIO_4_ and Al/CuO/70% KIO_4_ are 107 times and 152 times higher than that of Al/CuO, respectively. Similarly, the pressurization rate of Al/CuO/30% KClO_4_ is three and two orders of magnitude higher than that of Al/CuO and Al/KClO_4_. Thus, the reactivity of Al/CuO can be significantly improved and tailored by the addition of different contents of potassium oxysalts.

The reason why the combustion performance of Al/CuO is significantly enhanced by the addition of KIO_4_ and KClO_4_ can be speculated that the reaction between Al/KIO_4_ and Al/KClO_4_ starting at a low temperature is helpful to ignite the ternary composite materials. Meanwhile, it is speculated that CuO can accelerate the decomposition of KIO_4_ and KClO_4_ to release O_2_, leading to the reaction of Al with a large amount of gas. The higher flame temperature of Al/KIO_4_ or Al/KClO_4_ improves the pressure of the combustion cell and promotes the decomposition of the oxidizers further, thus more oxygen is produced, the oxidation of Al is further accelerated and the reaction rate is accelerated [19,27,28]. All these factors occur at the same time and have synergistic effects on enhancing the combustion performance of Al/CuO. In order to verify and quantify the effect of CuO on the thermal decomposition of KIO_4_ and KClO_4_, the TG analysis and DSC analysis of CuO/xKIO_4_ and CuO/xKClO_4_ were carried out in the later section.

### 3.4. Thermal Analysis

TGA/DSC were used to study the thermal decomposition reaction process of CuO/xKIO_4_ and CuO/xKClO_4_ at a low heating rate. Figure 4a demonstrates that KIO_4_ decomposes and releases O_2_ in two stages, which is consistent with the reported studies [21]. In the first stage, KIO_4_ decomposes exothermically into KIO_3_ and O_2_ with a peak temperature of the O_2_ release at ~343 °C. In the second stage, KIO_3_ starts the decomposition at ~535 °C with an endothermic peak at ~552 °C. Figure 4a reveals that the addition of CuO almost has no effect on the decomposition of KIO_4_ during the first stage with exothermic peak temperatures ranging from ~336 °C to ~339 °C. However, with the addition of CuO, the initiation and the endothermic peak of the second-stage decomposition of KIO_4_ advance by ~100 °C and ~90 °C compared with pure KIO_4_, respectively.

The thermal decomposition of KClO_4_ is shown in Figure 4b, KClO_4_ decomposes exothermically into KCl and O_2_ at ~595 °C with an exothermic peak at ~631 °C [29]. The endothermic peak at ~303 °C observed in DSC curves is the crystal transformation of KClO_4_, and the endothermic peak at ~609 °C is caused by the melting of KClO_4_. The addition of CuO has basically no influence on the crystal transformation of KClO_4_, which is ~300 °C. However, the addition of CuO advanced the melting and thermal decomposition of KClO_4_ significantly, resulting in the disappearance of endothermic peak of the melting and thermal decomposition at a lower temperature. The peak temperature of the thermal decomposition reaction of CuO/xKClO_4_ is ~500 °C, which is ~130 °C lower than that of pure KClO_4_. Moreover, the initial decomposition temperatures of KClO_4_ with different contents of CuO are between ~340 °C to ~390 °C, which are ~200 °C to ~250 °C lower than that of pure KClO_4_. This can be attributed to the fact that CuO, as a p-type transition metal oxide, promotes the electron transfer process during the decomposition of the oxidant [30,31,32].

Therefore, the addition of CuO has a significant effect on the thermal decomposition reaction process of KIO_4_ and KClO_4_, which makes the thermal decomposition reaction process earlier significantly. Meanwhile, the initial temperature and peak temperature of the oxygen releasing are greatly reduced, leading to the rapid reaction of Al with a large amount of gas through the pathways of the oxide shell, which may cause the internal molten Al to continuously absorb heat and expand, accelerating the rupture of the oxide layer, and further promoting the diffusion of Al core and oxidation of Al, thus enhancing the combustion performance of the composites [19,28,33,34].

### 3.5. Ignition Characterization

The combustion process of Al/CuO/xKIO_4_ and Al/CuO/xKClO_4_ composites has been recorded via high-speed camera. As shown in Figure 5, with the existence of KIO_4_ and KClO_4_ in the ternary systems, the initiation time of the reaction is earlier than that of Al/CuO.

Figure 5a demonstrates that the ignition delay time of the Al/CuO/xKIO_4_ ternary composites is shorter than that of the binary composites. Compared with Al/CuO (~1.28 ms), the ignition delay time of the composite with 50% KIO_4_ is shortened by ~0.38 ms. The ignition delay time of Al/CuO/70% KIO_4_ (~0.88 ms) is shorter than that of Al/KIO_4_ by ~0.32 ms. Among the Al/CuO/KIO_4_ composites, the ignition delay time of Al/CuO/30% KIO_4_ is the shortest, ~0.76 ms, which is ~0.52 ms and ~0.44 ms shorter than Al/CuO and Al/KIO_4_, respectively. As for the Al/CuO/xKClO_4_ systems, it is shown in Figure 5b that the ignition of composites with 30% KClO_4_ and 50% KClO_4_ occurs ~0.08 ms and ~0.14 ms earlier compared to Al/CuO. The ignition delay time of Al/CuO/70% KClO_4_ is the shortest, ~1.08 ms, which is ~0.20 ms and ~0.16 ms earlier than that of Al/CuO and Al/KClO_4_, respectively.

Herein, the ignition delay time of Al/CuO/xKIO_4_ and Al/CuO/xKClO_4_ ternary composites is shorter than that of that of both Al/CuO and Al/potassium oxysalts binary systems. It is reported that in the ternary thermites systems [35], the initiation of the whole system is triggered by the reaction between fuel and oxidizer that can be ignited at a lower temperature. Since the initiation temperature of Al/KIO_4_ and Al/KClO_4_ reaction is lower than that of Al/CuO [27,28], the ignition delay time of thermites composing of Al and potassium oxysalts is shorter than Al/CuO. We propose that the ignition of the ternary systems is initiated by the reaction of Al and potassium oxysalts. Combining the fact that the addition of CuO can advance the oxygen release of KIO_4_ and KClO_4_, the reaction between Al and potassium oxysalts in ternary systems can initiate earlier than Al/KIO_4_ and Al/KClO_4_. Therefore, the application of CuO/potassium oxysalts binary oxidizer system results in a synergetic effect including a lower ignition temperature and earlier oxygen release of oxidizers, thus significantly shortening the ignition delay of ternary thermites.

### 3.6. Heat Release Results

Assuming that the oxidant reacts completely with the fuel, the theoretical heat release of the composites is calculated based on the theoretical enthalpy of Al/CuO, Al/KIO_4_ and Al/KClO_4_ as shown in Equations (1)–(3), and their molar percentage in the ternary composites according to Hess’s law [26]. Figure 6 shows that with the increase of the content of KIO_4_ and KClO_4_, the heat release of the composites increases monotonously, which can be attributed to the reason that the theoretical enthalpy of Al/KIO_4_ and Al/KClO_4_ is higher than that of Al/CuO according to the Equations (1)–(3). The heat release of Al/CuO/30% KIO_4_ is 4958 ± 111 J g^−1^ (Figure 6a), which is ~1200 J g^−1^ higher than that of Al/CuO. Figure 6b demonstrates that the heat release of Al/CuO/50% KClO_4_ (6926 ± 150 J g^−1^) is ~3200 J g^−1^ higher than that of Al/CuO, which has been increased significantly. In addition, the experimental heat release of the composites is less than the theoretical heat release because the composites cannot react completely during the experiment, which can be confirmed by the XRD characterization of the combustion products (Appendix A). A more detailed description can be seen in the Appendix A.

The reaction completeness of composite system is obtained by calculating the ratio of the experimental calorific values to the theoretical ones. For the systems of Al/CuO/xKIO_4_, the reaction completeness of the ternary composite is higher than that of the binary systems. The highest reaction completeness of 96.1% for Al/CuO/xKIO_4_ ternary systems is achieved by Al/CuO/50% KIO_4_, which is 4.8% higher than that of Al/CuO. The reaction completeness of Al/CuO/xKClO_4_ is higher than that of Al/KClO_4_, and the reaction completeness of Al/CuO/30% KClO_4_ is 82.1%, which is 6.8% higher than that of Al/KClO4. It is presumed that as a transition metal oxidizer, CuO can promote the combustion of Al/KIO_4_ and Al/KClO_4_ as a catalyst. The relatively lower reaction completeness of Al/CuO/70% KClO_4_ might be caused by the lower ratio of CuO and the weakening catalysis of CuO.

It is reported that when there is more released gas during the combustion process of MIC, higher combustion efficiency could be achieved due to less aggregation of nAl [36]. Consistent with the results from this work, the higher peak pressure obtained from Al/CuO/KIO_4_ and Al/CuO/KClO_4_ composites (Figure 3a) means the amount of gas products is larger, which is corresponding to the higher reaction completeness of the ternary composites in Figure 6.

## 4. Conclusions

In this study, the binary oxidizers composed of potassium oxysalts and CuO were applied in nano Al-based MIC, and their energetic performance has been tested. The pressure cell tests show that the pressure peak and pressurization rate of Al/CuO/xKIO_4_ and Al/CuO/xKClO_4_ ternary composites are significantly higher than that of Al/CuO, Al/KIO_4_ and Al/KClO_4_ binary composites. It is speculated that the synergistic effect of CuO/KIO_4_ and CuO/KClO_4_ binary oxidizers make the combustion performance of the ternary composites better than that of the binary composites. In other words, the reaction of Al/KIO_4_ and Al/KClO_4_ at a lower temperature is helpful to ignite the ternary composite. Meanwhile, CuO promotes the decomposition of KIO_4_ and KClO_4_ to release oxygen, which makes Al react quickly with a large amount of gas, accelerates the rupture of the oxide layer, and further promotes the diffusion of Al core and oxidation of Al, thus enhancing the combustion performance of the composites. The results of TG-DSC show that CuO significantly advances the initial temperature and peak temperature of KIO_4_ oxygen release in the second-step decomposition by ~100 °C and ~90 °C, respectively, and decreases the initial temperature and peak temperature of KClO_4_ oxygen release by ~200 °C and ~130 °C, respectively. Moreover, the results of ignition tests and heat release measurement show that the ignition delay time of the ternary composites is shorter than that of Al/CuO. The addition of KIO_4_ and KClO_4_ can tailor the reaction completeness and heat release of the composites due to more released gas by the reaction between Al and potassium oxysalts. Therefore, KIO_4_ and KClO_4_ are promising additives for Al/CuO to tune and promote the combustion performance.

## Figures and Tables

**Figure 1 nanomaterials-11-03366-f001:**
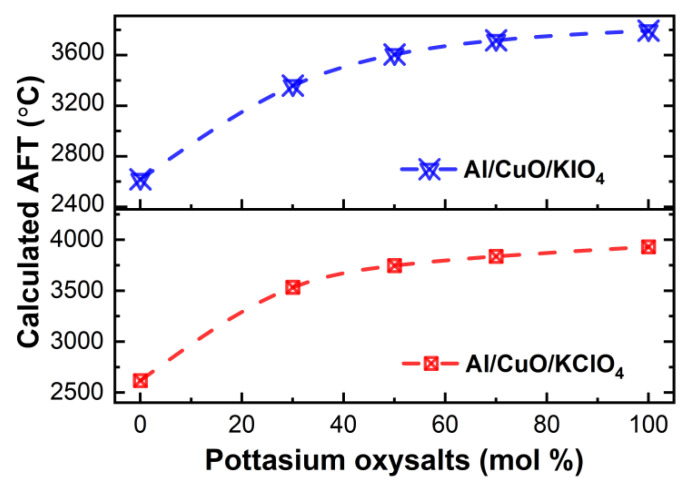
The calculated AFT of Al/CuO/potassium oxysalts systems by REAL.

**Figure 2 nanomaterials-11-03366-f002:**
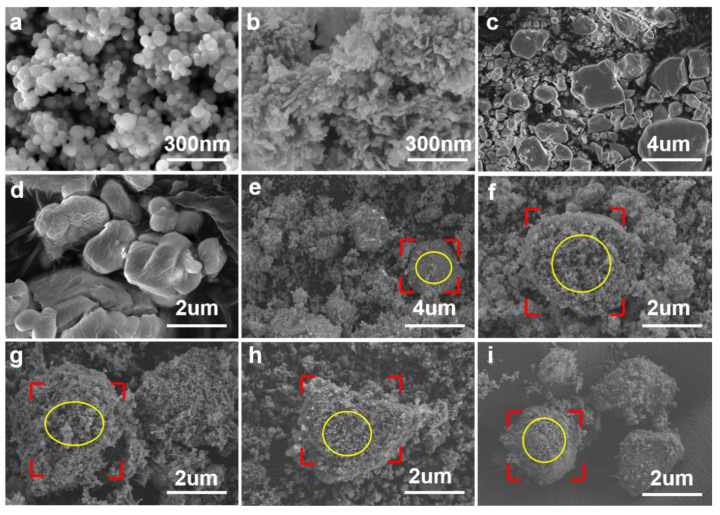
SEM images of samples: (**a**) Al, (**b**) CuO, (**c**) KIO_4_, (**d**) KClO_4_, (**e**) Al/CuO/30%KIO_4_, (**f**) Al/CuO/50%KIO_4_, (**g**) Al/KIO_4_, (**h**) Al/CuO/50%KClO_4_, (**i**) Al/KClO_4_.

**Figure 3 nanomaterials-11-03366-f003:**
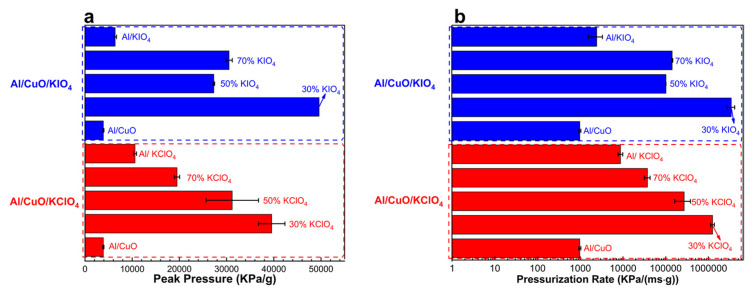
The measured peak pressure (**a**) and pressurization rate (**b**) of Al/CuO/potassium oxysalts systems. Note: “70%” means “Al/CuO/70% potassium oxysalts”.

**Figure 4 nanomaterials-11-03366-f004:**
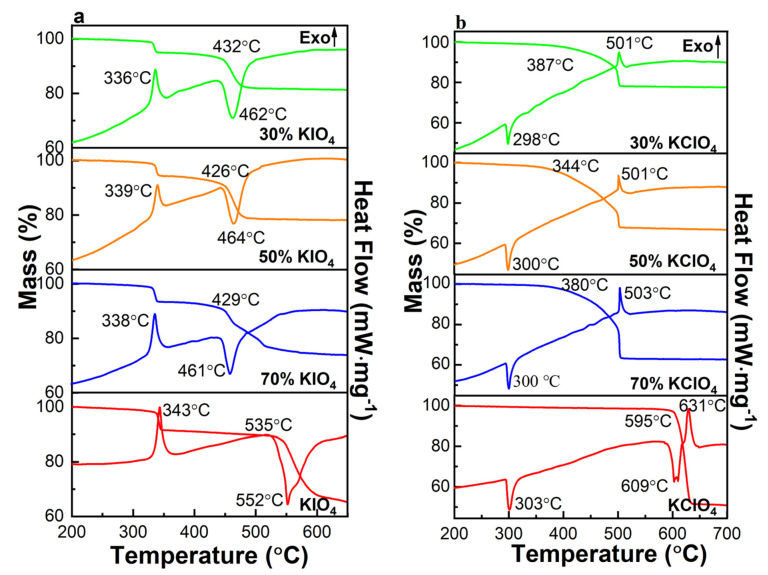
DSC and TG curves of (**a**) CuO/xKIO_4_ and (**b**) CuO/xKClO_4_.

**Figure 5 nanomaterials-11-03366-f005:**
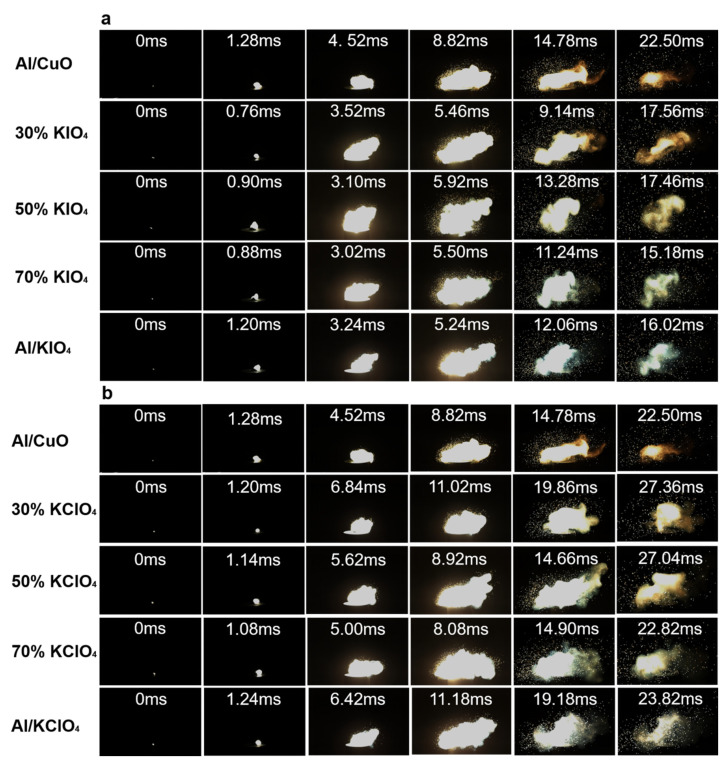
The combustion snapshots of (**a**) Al/CuO/xKIO_4_ and (**b**) Al/CuO/xKClO_4_ systems-based composites.

**Figure 6 nanomaterials-11-03366-f006:**
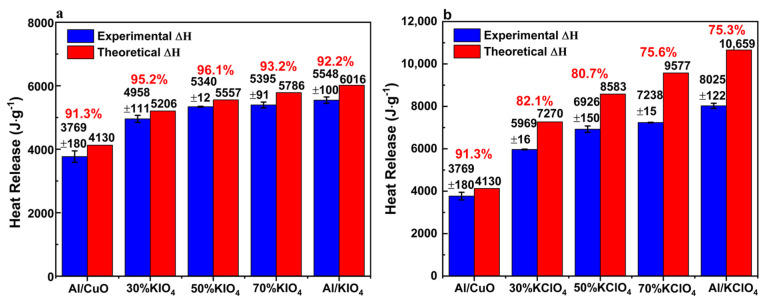
The experimental, theoretical enthalpy, and corresponding combustion efficiency of (**a**) Al/CuO/xKIO_4_ and (**b**) Al/CuO/xKClO_4_ composites. Note: ‘‘30% KIO_4_” in Al/CuO/xKIO_4_ system means Al/CuO/30%KIO_4_.

**Table 1 nanomaterials-11-03366-t001:** Stoichiometric thermites formulation (Al/CuO/x potassium oxysalts).

Thermites	Al (wt%)	CuO (wt%)	Potassium Oxysalts (wt%)
Al/CuO	27.8	72.2	0.0
Al/CuO/30% KIO_4_	31.7	30.5	37.9
Al/CuO/50% KIO_4_	32.9	17.2	49.8
Al/CuO/70% KIO_4_	33.9	8.5	57.6
Al/KIO_4_	34.6	0.0	65.4
Al/CuO/30% KClO_4_	37.4	35.9	26.7
Al/CuO/50% KClO_4_	41.0	21.5	37.4
Al/CuO/70% KClO_4_	43.8	11.1	45.1
Al/KClO_4_	46.8	0.0	53.2

## Data Availability

Data is contained within the article or Appendix A.

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
