# Peer review of "Synergetic Effect of Potassium Oxysalts on Combustion and Ignition of Al/CuO Composites"

_nanomaterials, 2021, doi:10.3390/nano11123366_

Round 1

Reviewer 1 Report

The authors, through extensive characterizations have demonstrated the positive attributes of adding potassium oxysalts to Al/CuO nanothermites on the combustion and ignition properties. The mechanism enhanced combustion performance has been speculated to the “reaction between Al/KIO4 and Al/KClO4 starting at a low temperature is helpful to ignite the ternary composite materials” This statement supports the ignition property but not the combustion. Combustion enhancement should be related to the improved oxidation of Al by the breakage of the alumina shell in the presence of the CuO and potassium oxysalt oxidizer. A more detailed explanation in this respect with a proper review of associated literature is required to support the combustion enhancement.

The reported peak pressure and pressurization rate should be normalized with the amount of sample used. The basis of the calculation of the theoretical heat release of the composites presented in section 3.6 should be indicated in the manuscript.

This material system has been extensively investigated in technical literature [ eg: F.Yang et.al; scientific reports 7, 3730 (2017); C.Wu et.al;  Adv. Funct. Mater. 2012, 22, 78–85]. It is instructive to compare the results obtained in this manuscript with that reported in the literature.

Reviewer 2 Report

The authors investigated the synergetic effects of potassium oxysalts on combustion and ignition of nano Al and nano CuO composites. They found the reaction of Al/KIO4 and Al/KClO4 at a lower temperature is helpful to ignite the ternary composite, and they obtained a significant result that KIO4 and KClO4 are promising additives for Al/CuO to tune and promote the combustion performance. As a result of having been provided in this article, I think that it is very significant.

1. p.3 L109-116 : In section 2-5, the authors described about the heat release measurement. Show the details of this measurement by means of a figure to make it plain to readers.
2. p.6 L201-205 : “The peak temperature of the thermal decomposition reaction of CuO/xKClO4 is ~500 °C, which is ~130 °C lower than that of pure KClO4. Moreover, the initial decomposition temperatures of KClO4 with different contents of CuO are between ~340 ℃ to ~390 °C, which are ~200 °C to ~250 °C lower than that of pure KClO4.”  Explain this fact scientifically. 
3.  p.8 Figure 5 : Answer how you confirmed that the composites were completely burned.
4. p.9 Figure 6 (a), (b) : Explain the result that the experimental results were smaller than the theoretical results. 
